# Flexible Goal Programming for Supporting Lake Karla's (Greece) Sustainable Operation

**Mike Spiliotis** [1,*] , **Dionissis Latinopoulos** [1] , **Lampros Vasiliades** [2] , **Kyriakos Rafailidis** [1] , **Eleni Koutsokera** [1] **and Ifigenia Kagalou** [1]

1   Department of Civil Engineering, School of Engineering, Democritus University of Thrace, 67100 Xanthi, Greece; dlatinop@civil.duth.gr (D.L.); kiriakosraf@hotmail.com (K.R.); elenkout7@civil.duth.gr (E.K.); ikagkalo@civil.duth.gr (I.K.)
2   Department of Civil Engineering, School of Engineering, University of Thessaly, 38334 Volos, Greece; lvassil@civ.uth.gr
*   Correspondence: mspiliot@civil.duth.gr; Tel.: +30-25-4107-9613

**Abstract:** Sustainable management is a prerequisite for a lake to provide a range of ecosystem services. The prioritization of needs is a difficult task, especially when the needs are in conflict and threaten water security. Lake Karla, situated in the Thessaly plain, Greece, was decimated in 1957–1962; due to environmental impacts, it was later refilled as a multipurpose reservoir with high ecological significance. The research objective is to achieve a compromise with respect to both the economic benefits derived from agricultural water use and environmental protection based on the minimum intersection. For this purpose, first, new managerial practices are introduced. Second, the ideas are quantified based on the hydrological budget, and these are used as input for flexible (fuzzy) programming. Under hypotheses about the acceptable range, the (flexible) fuzzy programming is identical with the MINMAX goal programming model, although the weights are not used directly in the first case. An understandable compromise (the maximum economic benefit from irrigation areas and the minimization of water retention time) is achieved, and the values of the membership functions can be used to verify the solution. The proposed solution leads to a quantitative proposition, incorporating new findings from modeling the recent real operation of the reservoir.

**Keywords:** Karla Lake; lake restoration; fuzzy sets and logic; flexible programming; goal programming

## 1. Introduction

Water-related ecosystems provide multiple benefits and services to society, making them essential for reaching several Sustainable Development Goals (SDGs) posed by the United Nations [1]. Ensuring sustainable water systems requires a fundamental change in how water is used by various sectors and valued by society. For sectors that disproportionately affect water sustainability (e.g., urban water management, agriculture, energy), the EU Adaptation Strategy [2] suggests more transformational approaches to safeguard resilient water systems. These include, for example, changes towards sustainable agricultural systems, green infrastructure, nature-based solutions, stakeholder participation and raising awareness [3–6].

Regarding freshwaters, it has been recognized that their continued deterioration is not sustainable and that less modified freshwater ecosystems provide significant economic and social benefits to society. The Water Framework Directive (WFD) [7] included a holistic common approach for all Member States (MS). Despite ongoing efforts by the MS for "good ecological status" as the main WFD objective, good ecological status is currently not achieved in 59% of rivers and 48% of lakes [8]. In Southern Europe, the most probable driver for this deterioration is agriculture, since it accounts for more than half of the total national abstraction, more than 80% in some regions, reaching unsustainable levels [8,9]. Among Mediterranean water bodies, shallow lakes and wetlands are the most

threatened ecosystems and are of particular importance since they are subject to multiple pressures [10–12], being sensitive to drought due to the low annual precipitation and high evaporation [13]. Thus, management of Mediterranean basins becomes more urgent, where sustainable water management is crucial not only for negating the multiple pressure effects that the water bodies are subject to, but also for ensuring the multiple related ecosystem services (potable, industrial, and agricultural water provisioning, food processing, and ecosystem preservation). Reviewing the relevant management studies, they progressively shift from specific water quality problems to catchment processes at a much broader level. This also reflects the tendency toward devising integrated watershed models in which the water quality is a module in the entire catchment model [14]. Each hydrological catchment is subject to overarching water balance [15], coupled with different water uses, conditions, management scenarios and socio-economic systems [16]. Thus, it is a strategic challenge to maximize a society's benefits from the various possible uses of available water resources, while ensuring that basic human needs are met and the environment is protected.

Greece is a very specific case, since previous comparative multi-catchment studies have shown that hydro-climatic changes and associated freshwater interactions with the atmosphere are subject to particularly large uncertainties [17], creating a particular interest in freshwater conditions and changes. The Thessaly plain, which includes the main basin of Pinios river and the sub-basin of lake Karla, is a region in central Greece facing significant water resource issues. It is an intensely cultivated region where extensive agriculture with water-demanding crops, such as cotton and maize, has led to a remarkable water demand increase, which is usually fulfilled by the overexploitation of groundwater [18,19]. The peculiarity of the lake lies in the fact that Karla used to be a natural shallow lake, one of the most important Greek shallow lakes. In 1962, it was dried out for agricultural and health-related (malaria) purposes. In 2010, after more than 30 years of groundwater over-abstraction and fertilizing of water-intensive monocultures, it was re- constructed. Its reconstruction was deemed as one of the most important restoration projects in Europe, and its success was considered of high importance by the European Union, as it offered multiple services, i.e., social, economic and ecological sustainable development to the region and not just creating a new reservoir.

Therefore, the utility of multicriteria methods is obvious in water resource management as compared to the monocriterion approach (e.g., cost-benefit analysis). The International Society on Multiple Criteria Decision Making [20] defines MCDM as "the study of methods and procedures by which multiple and conflicting criteria can be incorporated into the decision process". Sustainable water management is a complex challenge, where various factors need to be involved. During the last two decades, fuzzy logic has gained more and more ground [21]. The use of such methods, especially in decision making, supports sustainable management and enhances objectivity [22–24]. The different dimensions of the criteria, the conflict among them and other various uncertainties create the need for a multicriteria approach. In addition, the existing knowledge of experts can be incorporated by using fuzzy-oriented multicriteria methods.

Using the multicriteria optimization problem is a way to build a new, integrated solution [20]. The application of such a method could be enhance management when the hydrological regime is dramatically changed and hence a redesign of the work is required [25]. Although Zimmerman's approach uses both objectives and constraints as fuzzy [26], in many studies, only the objectives are considered as fuzzy. The use of fuzzy methods instead of a conventional approach has the following advantages [27]: (1) fuzzy uncertainties can be directly included, (2) the variation or fuzziness of the decision maker's aspirational level in the model can be taken into account, and (3) each objective or goal has its own independent membership function.

The fuzzy optimization could be proposed as multicriteria optimization method. For this purpose, a fuzzy optimization method can be used, where the use of the minimum (min) intersection ensures a common acceptable finally crisp solution. Therefore, even if an ideal solution of this problem, i.e., a solution that maximizes all objectives simultaneously,

does not exist in general, by using the concept of membership function, a common acceptable common degree of aspiration level among the membership functions can be achieved. Hence, the use of the minimum intersection leads to an understandable compromise solution, with the maximum common acceptable degree of satisfaction for all the membership functions. In fact, fuzzy multicriteria optimization is an interactive process where the information can be utilized by means of the fuzzy membership function. Fuzziness, imitating the human way of thinking, can provide an understandable solution without ambiguous considerations [28], such as the direct choice of the weights or the way of aggregation of several objective functions among other parameters [29]. An advantage of fuzzy flexible programming is that the decision maker can model his or her problem based on his or her current state of information and modulate the fuzzy procedure accordingly [30]. Especially when there is a lack of information, the simple flexible fuzzy programming with uncertainty only in the right hand of the inequalities could be of use. This type of fuzzy programming is named flexible programming, and it is strongly related to conventional goal programming. Under some hypotheses about the acceptable range, the (flexible) fuzzy programming can be identical to the MINMAX goal programming (GP) model (presented in Section 4.2 here), although the weights cannot be used when fuzzy flexible programming is applied.

The article deals with the restored Lake Karla, which is a complicated multi-purpose operating reservoir. The reservoir operation is based on river discharges diverted from the adjacent Pinios River and on surface runoff from the Lake Karla watershed. Based on recent findings from the reservoir operation, extensive alterations associated with land use changes, hydrological flow modifications, over-enrichment of chemicals, and inappropriate management of biological resources are observed [31,32]. Furthermore, several subprojects associated with the water supply from the Lake for agricultural and urban water uses are not yet completed [33]. Hence, in this study, the initial design and operation of the reservoir is updated and validated using new elements and information derived from the real reservoir operation [33]. Based on the above findings, it is concluded that because of the intensive use of the groundwater and the existence of several small dams, and due to many arbitrary local water abstractions, the local basin has a small contribution to the reservoir water balance, which is dramatically different from the initial planning studies. Starting from this new finding, the proposed solution incorporates some new ideas as to the use of winter crops in order to reduce the retention time. These new ideas are quantified based on the proposed fuzzy GP, with two objectives. Finally, we highlight that, based on the proposed (fuzzy) multicriteria approach, an analytical quantitative solution is achieved rather than general strategies.

The paper presents the following, in order: i) a description of the study area, ii) the new ideas and challenges in this study, iii) the hydrological and mathematical tools used in order to address the challenges and model the reservoir operation under a new realistic prism, where flexible fuzzy and GP models (and their relation) are presented, iv) results, and v) discussion and conclusions.

## 2. Study Area and Reconstruction Project

Lake Karla ($39°31'60.00''$ N, $22°41'60.00''$ E, lake area: ~38 km$^2$) is under multiple pressures, mainly agriculture, uncontrolled farming, water abstraction, hydrology regulation, and low catchment inflows, resulting in its eutrophication.

Historically, before its drying, surface runoff from the watershed and floodwaters of the Pinios River supplied the lake with large quantities of freshwater. Its surface area fluctuated between 40 km$^2$ and 180 km$^2$. The structure and function of Lake Karla was intimately linked with the Pinios River. The river occasionally overflowed, and floodwaters rich in oxygen and nutrients drained into Karla. Much of the surrounding farmland was inundated when floodwaters were held in the lake, but today, the river is leveled for flood protection reasons. The climate is typical continental. The Thessaly, and especially the Lake Karla watershed, experienced severe, extreme, and persistent droughts from the mid to late 1970s, from the late 1980s to the early 1990s, and during the early 2000s [18]. The lake's

sedimentary aquifer occupies the largest part of its plain, with an extent of 500 km². It is being over-exploited, covering both the irrigation needs of the cultivated areas and the supply water needs of the settlements [33].

The re-construction project has been described in detail in many recent studies [33–35]. Based on these, the recreated Lake Karla watershed has a drainage area of 1171 km² (Figure 1). In brief, the restoration plan proposes the creation of a reservoir in the lowest depression plain of the former Lake Karla, which will occupy a maximum area of about 38 km², through the construction of two embankments. Inflows come from two main ditches that transfer the flood runoff of the Pinios to the reservoir. According to the approved River Basin Management Plan (RBMP) for the Thessaly Water District [36], a significant amount of water could be abstracted from the Pinios during the high-flow winter period to ensure sustainability the of Karla reservoir. Using water balance analysis, the annual water amount diverted from the Pinios flood that flows during the winter season is approximately 100 hm³ (YPEKA), justified by the estimated water demands for irrigation. Furthermore, four collector channels concentrate the surface runoff from the sub-basin and pump it into the reservoir (Figure 1). Thus, the maximum allowable volume of the reservoir reaches up the 180 hm³, but only 60 hm³ is available to fulfill the irrigation needs of the surrounding agriculture because of environmental restraints [37]. Having presented the project and construction, we will now address the gaps and discuss the local problems.

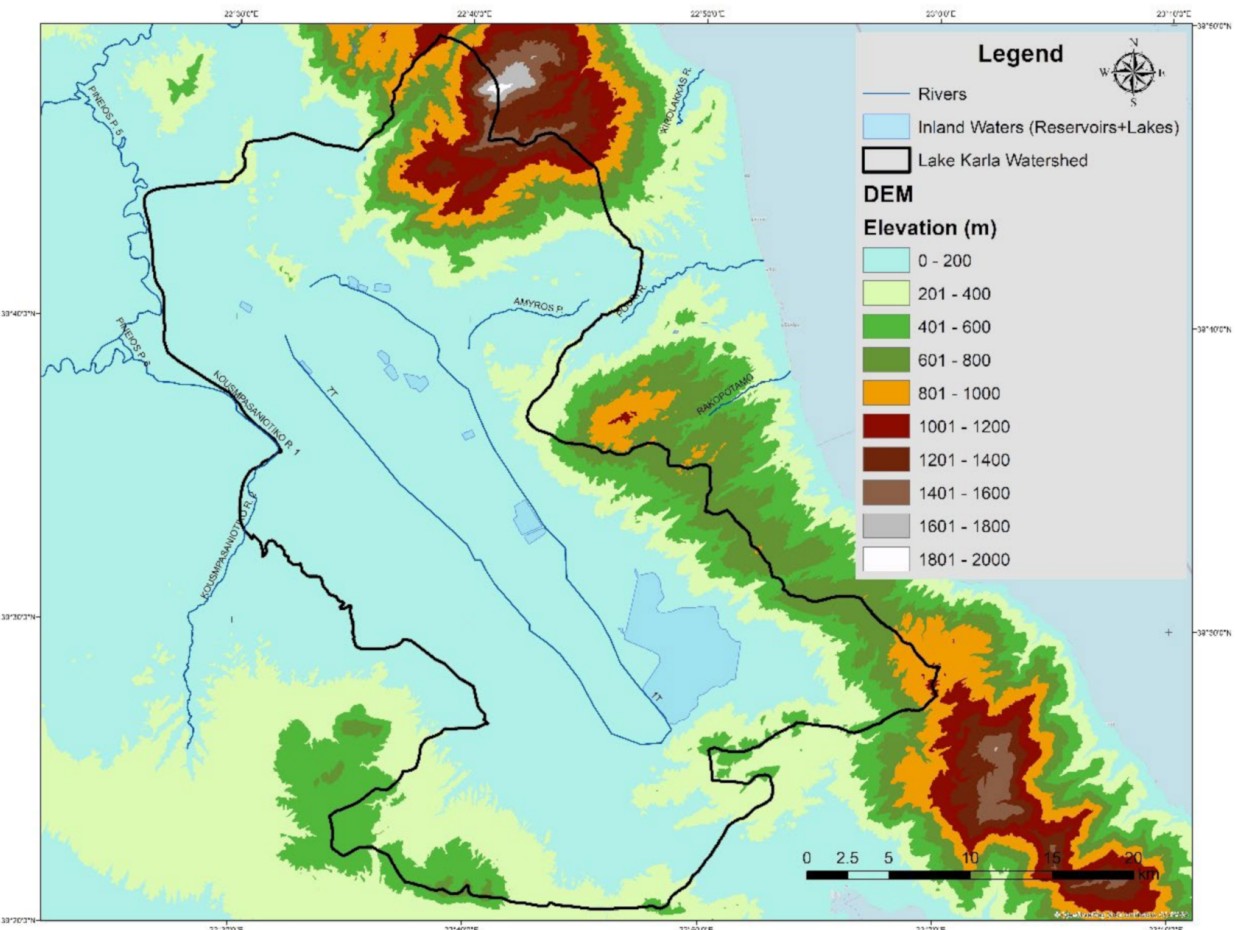

**Figure 1.** Complete hydrographic network of Lake Karla watershed using a Digital Elevation Model (DEM) with the location of new reservoir, ditches, and water collection points.

## 3. Setting the Scene

### 3.1. Acknowledging the Gaps

The newly created reservoir faces several issues in not complying with local needs, which are supposedly addressed by the reconstruction plan.

- The reservoir suffers from very low inflows besides precipitation, and coupled with the fact that the irrigation abstraction takes place during the summer period, the water retention time is close to infinity [38].
- Water ecological quality has been characterized as less than poor since 2014, when the national monitoring program for WFD was started [36,39,40]. This is related to the previous point, since it has been acknowledged for more than 30 years that high water retention has significant negative effects on water quality [41–43].
- The soils are degraded and high in salts, lowering the agricultural productivity, which is in contradiction with local needs [4,44].
- The aim of supplying the urban area of Volos city with drinking water through the aquifer enrichment needs more time for fulfilment.
- A delay in construction works and the non-complete reservoir operation result in stakeholder fatigue by not providing the entirety of the project's benefits.

### 3.2. Setting the Goals

The purpose of the article is to find a compromise between environmental restoration and covering societal needs concerning agricultural production and water allocation. To come up with such a solution, riverine inflows, Karla reservoir operation, and groundwater tables must be taken into account in a holistic and sustainable manner. Fuzzy application can serve this goal by supporting decision making in favor of the aquatic ecosystem's sustainability. The conflict of these two goals, and the allowable tolerance of each goal, insert the fuzziness in the problem. The solution is achieved by adopting the minimum intersection that leads to a common satisfaction level for both goals. The proposed fuzzy approach is a type of Goal Programming method (GP).

The authors' suggestion for an ecological/agricultural solution is founded on three main admissions and constraints, encompassing both expert and stakeholder opinions (lake Karla Management Body, Prefecture, Department of agricultural economy, and Region of Thessaly):

- The enhancement of winter crops as a practice in order to reduce the retention time in the Karla reservoir and to simultaneously improve the soil fertility. The proposed solution by the authors suggests winter crops to extend to one-third of the summer cultivated areas irrigated by the reservoir, adopting a basic CAP measure such as crop rotation (GAEC 8) [45].
- The use of the maximized limit for acceptable inflow, with 100 hm$^3$ as a constant, or augmentation of the predicted flood volume from the Pinios [36].
- An additional preset amount of water to be released monthly from the reservoir for multiple purposes, ensuring the sustainability of the hydrological regime [39].

The above challenges must be included within the proposed quantitative model as parts of the proposed solutions. To address them properly, first, an adapted hydrological budget must be computed. The main innovation of this work is the use of a fuzzy decision scheme, which is additionally coupled with the simulation model. The fuzzy programming is applied to quantify this hypothesis by finding a balance as a compromise solution between water retention time minimization in favor of the water quality and the economic benefits from agricultural practice maximization. Since the above management targets create a conflict for the water use, fuzzy flexible programming [46] is implemented in order to achieve a compromise for ensuring Karla reservoir sustainability. The proposed model can provide an understandable decision-making scheme without the direct use of the weights.

## 4. Methodological Approach

### 4.1. Hydrological Budget

Monthly precipitation and temperature data of 12 uniform distributed meteorological stations across the watershed were available for the period October 1960 to September 2002 (Figure 2). Hence, monthly areal precipitation and temperature of the basin were estimated by the Thiessen polygon method, modified by the precipitation/temperature gradient, using the stations which are within or in the vicinity of the watershed (Figure 2). Potential evapotranspiration was estimated using the Thornthwaite method, based on areal monthly temperature values. Figure 2 also shows the Lake Karla plain and mountainous areas, based on an elevation threshold of 200 m, which demonstrates that the Lake Karla aquifer is situated on the plain area.

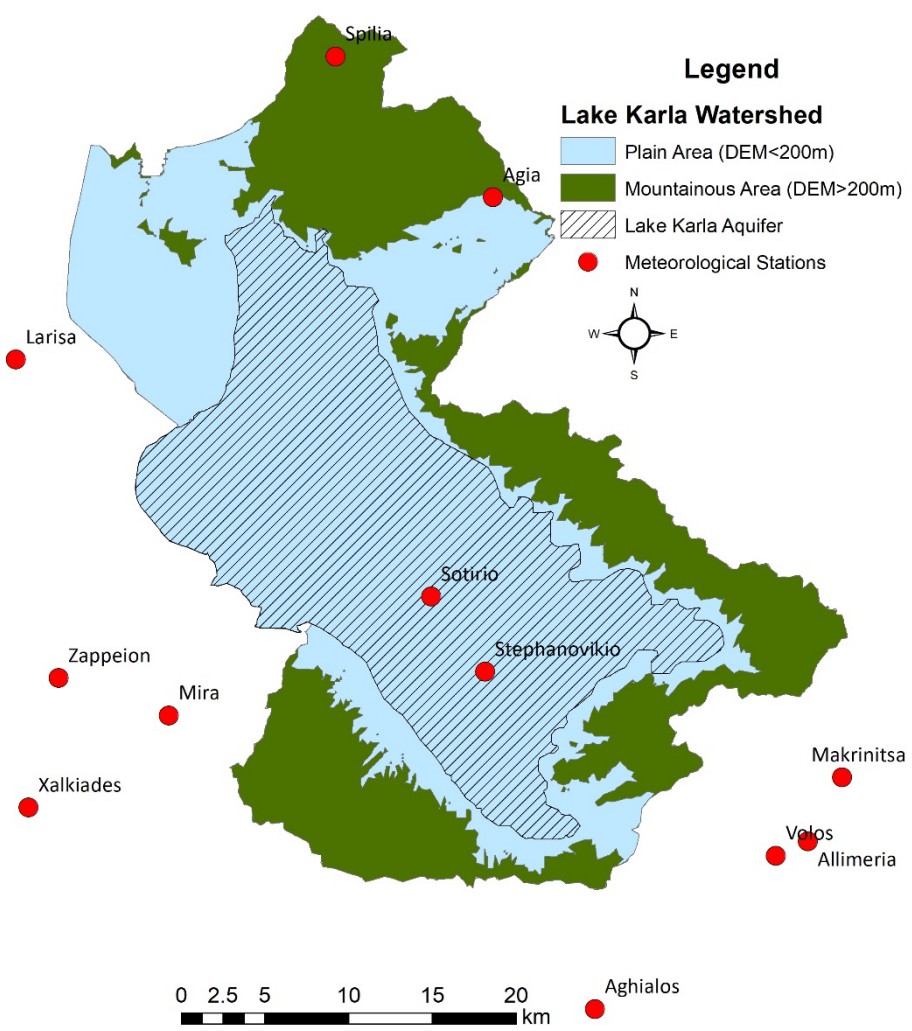

**Figure 2.** Location of meteorological stations in the study area.

The analysis is based on the operation of the reservoir during 2018, where an analytical reservoir water budget was estimated based on the Hydromentor modelling system [47]. The Hydromentor system is a suite of coupled mathematical models for assessing water balance components and their contribution to reservoir operation due to different operational and climate change scenarios in semi-arid watersheds. Further details can be found in recent papers [35,47]. Based on these modelling results, almost 34.4 hm$^3$ are provided on a yearly basis by the Pinios River, which is the main water source for the system. Because of the intensive use of the groundwater and the existence of several small dams/reservoirs, and due to many arbitrary local water abstractions, the local basin

has a small contribution to the reservoir water balance (almost 6.8 hm$^3$). Based on this finding, the Giakoumakis et al. [48] monthly hydrological model is calibrated to provide the contribution of the local basin based on the mean climatological conditions. This model is selected due to its simplicity (few parameters) and ability to deal with a possible lack of data.

The agricultural water requirements of the summer crops are calculated by the authors based on the current allocation, and the proposed winter crop requirements are based on the Blaney-Griddle method, taking into account the effective rainfall [49]. Summer crop requirements were calculated based on localized data and coefficients [50].

### 4.2. Fuzzy Flexible Programming

The aforementioned two goals are in conflict, since the practice with extended irrigated areas leads to high retention time due to the different seasonality of the irrigation needs and the river flood inflows. Therefore, the scope of the analysis is to produce a common acceptable solution with high benefit from the irrigated areas and simultaneously with low retention time. These two goals are expressed by using the membership function. This can be achieved if the objective function takes values between the closed interval [0,1]:

$$\mu_G : X \to [0,\ 1], \tag{1}$$

where $\mu_g$ and X are the membership function and the general set, respectively.

To aggregate the adopted goal, Bellman and Zadeh [46] suggested the definition of the decision set based on the fuzzy intersection. In many applications, the minimum (min) intersection is used, coinciding with the intersection of the conventional logic. Therefore, the problem concludes in the following equation [26]:

$$\mu_D(\mathbf{x}) = \min\big(\mu_{G_1}(\mathbf{x}), \mu_{G_2}(\mathbf{x})\big), \tag{2}$$

where, $\mu_D(\mathbf{x})$ is the membership function of the decision space, and $\mathbf{x}$ is the vector of the decision variables. Therefore, the term $\mu_D(\mathbf{x})$ expresses the membership function of the intersection $G_1 \cap G_2$. In this application, the decision variables are the irrigated areas and the amount of water that can be constantly withdrawn for the various uses.

Since the decision maker should conclude in a crisp decision proposal, it seems more appropriate to select the vector that concludes to the maximum common degree of satisfaction for both fuzzified goals.

$$\mu_D(\mathbf{x}) = \min\big(\mu_{G_1}(\mathbf{x}), \mu_{G_2}(\mathbf{x})\big), \tag{3}$$

Hence, by using the auxiliary variable $\lambda$, the problem concludes in the following problem:

$$
\begin{aligned}
&\max \lambda \\
&\text{s.t.} \\
&\mu_{G_1}(\mathbf{x}) \geq \lambda, \\
&\mu_{G_2}(\mathbf{x}) \geq \lambda, \\
&\alpha \in [0,1] \\
&\mathbf{x} \in X \\
&\text{crisp constraints}
\end{aligned}
\tag{4}
$$

Since the crisp constraints are non-linear, the problem concludes to a nonlinear programming model. This approach is called the MINMAX approach, flexible programming, or fuzzy programming. An interesting point of view is that the use of the weights among the objective functions in this approach is avoided.

The main question to be addressed is the construction of the membership function of the goals.

The membership function, which expresses the goal of maximum cultivated areas, is a benefit-type membership function, dependent on the area of the cultivated areas. The

membership function, which expresses the goal of minimum high retention time, is a cost-type membership function, dependent on the area of index $\tau$:

$$\sum_{\iota=1}^{12}\left(\frac{V_i}{\sum\limits_{outflow} Q_i}\right) = \tau \qquad (5)$$

where $V_i$, $\sum\limits_{outflow} Q_i$ is the value of the water volume of the reservoir and the total amount of outflows of the lake during the month $i$.

By using linear membership functions, the key question is the selection of their thresholds, that is, the lowest acceptable level and the aspired level. A common practice is to use the values of the individual monocriterion solutions. At first, the multi-objective problem is solved for each objective separately. Let $\mathbf{x}^{(1)*}$, $\mathbf{x}^{(2)*}$ be the matrix of the optimal values achieved for each solution.

Thus, a pay-off matrix P is expressed as follows:

$$P = \begin{bmatrix} \tau\left(\mathbf{x}^{(1)*}\right), & \tau\left(\mathbf{x}^{(2)*}\right) \\ A_1\left(\mathbf{x}^{(1)*}\right), & A_1\left(\mathbf{x}^{(2)*}\right) \end{bmatrix}. \qquad (6)$$

The lowest acceptable (tolerance) levels and the aspired levels are selected from the pay-off matrix [29,51]. The matrix elements of the diagonal represent the *K* aspired levels for each objective. For instance, the maximization of the summer cultivated areas, $A_1$ (**x**), should be mathematically quantified as follows:

$$\mu_{A_1}(\mathbf{x}) = \begin{cases} 1 & \text{if } A_1(\mathbf{x}) \geq A_1\left(\mathbf{x}^{(2)*}\right) \\ 0 & \text{if } A_1(\mathbf{x}) \leq A_1\left(\mathbf{x}^{(1)*}\right) \\ \dfrac{A_1(\mathbf{x}) - A_1\left(\mathbf{x}^{(1)*}\right)}{A_1\left(\mathbf{x}^{(2)*}\right) - A_1\left(\mathbf{x}^{(1)*}\right)} & \text{if } A_1\left(\mathbf{x}^{(1)*}\right) \leq A_1(\mathbf{x}) \leq A_1\left(\mathbf{x}^{(2)*}\right) \end{cases} \qquad (7)$$

As presented in Table 1, three types of membership function are used in fuzzy (flexible) programming. In this article, the first two cases are used. The uncertainty of the aspiration level is strongly related to the weight of the conventional GP, which will be presented later.

It should be clarified that the GP initially starts with the third type of constraint, which can be interpreted linguistically as "almost equal". This constraint is very useful in the case that a river without a reservoir is studied (e.g., [29]). However, both the GP and the flexible programming can cover the case of "greater than (with a tolerance)" and "smaller than (with a tolerance)".

An interesting point of view is the connection between the conventional GP and the fuzzy programming. Hannan [52] studied the interconnection between the flexible programming and the conventional GP in the case of isosceles membership function (case three in Table 1). Yaghoobi and Tamiz [53] generalize the interconnection between the flexible programming and the conventional GP (Table 1) for several membership functions initially based on the min–max problem. In fact, the weights are used, but they can simply be interpreted based on the fuzziness of the aspiration level. As aforementioned, in this problem, the fuzziness is modulated based on the pay-off matrix. Therefore, in Table 1, $\lambda$ is the common level regarding the satisfaction of the fuzzy goals.

$$\left.\begin{array}{l} p = d^+ \cdot \Delta^+ \\ n = d^- \cdot \Delta^- \end{array}\right\}, \qquad (8)$$

where p and n are the exceedance and the shortage from the ideal values, $\Delta^+$, $\Delta^-$ are the corresponding acceptable ranges (based on the fuzziness), and $d_i^+$, $d_i^-$ are the corresponding dimensionless divergences from the ideal values.

Hence, based on Table 1, a linkage (and/or intersection) can be found between the flexible programming and the special case of the MINMAX (GP) [53]. Furthermore, based on this study, another linkage was proposed between the weighted GP and the flexible programming. So, for this case study, the examined problem becomes

$$
\begin{aligned}
&\min\left(w_2\frac{p}{\Delta^+} + w_1\frac{n}{\Delta^-}\right)\\
&\text{s.t. } \tau(\mathbf{x}) - p \le b,\\
&\quad A_1(\mathbf{x}) + n \ge b,\\
&\quad n, p \ge 0,\\
&\quad \frac{n}{\Delta^-} \le 1,\\
&\quad \frac{p}{\Delta^+} \le 1,\\
&\quad X \in C_s
\end{aligned}
\tag{9}
$$

Therefore, a weight for each objective function and the sum of the diversions are defined instead of the min intersection.

In general, GP includes the aforementioned MINMAX GP, the weighted GP, and the lexicographic approach, among others, as depicted in Figure 3. The fuzzy programming first consists of flexible programming, where there is an uncertainty on the right hand of the constraint (Table 1) and the solution is based on the membership function, while the other parameters remain crisp numbers. In the case that the uncertainty appears in the coefficients of the decision variables, the fuzzy programming is named possibilistic programming, which is selected for an ill-defined problem. For instance, the interpretation of the fuzzy inequalities is an open challenge [54]. An important case of the possibilistic programming is when the decision variables are fuzzy numbers [55].

**Table 1.** Correspondence between the flexible programming and the MINMAX GP.

| Type of Uncertainty | Membership Function (Simplification for Linear Case) | Correspondence with Goal Programming | Use in This Article |
|---|---|---|---|
| $A_1(\mathbf{x})\widetilde{\ge}b$ |  | $\max\lambda$ $A_1(\mathbf{x}) + n \ge b$ $\lambda + \dfrac{1}{\Delta^-}n \le 1$ $\lambda, n \ge 0$ $X \in C_S$ | Maximum cultivated area (as usual) |
| $\tau(\mathbf{x})\widetilde{\le}b$ |  | $\max\lambda$ $\tau(\mathbf{x}) - p \le b$ $\lambda + \dfrac{1}{\Delta^+}p \le 1$ $\lambda, p \ge 0$ $X \in C_S$ | Minimum index of retention time |
| $A_3(\mathbf{x})\widetilde{\cong}b$ |  | $\max\lambda$ $A_3(\mathbf{x}) + n - p = b$ $\lambda + \left(\dfrac{1}{\Delta^-}n\right) + \dfrac{1}{\Delta}p \le 1$ $\lambda, n, p \ge 0$ $X \in C_S$ | No use |

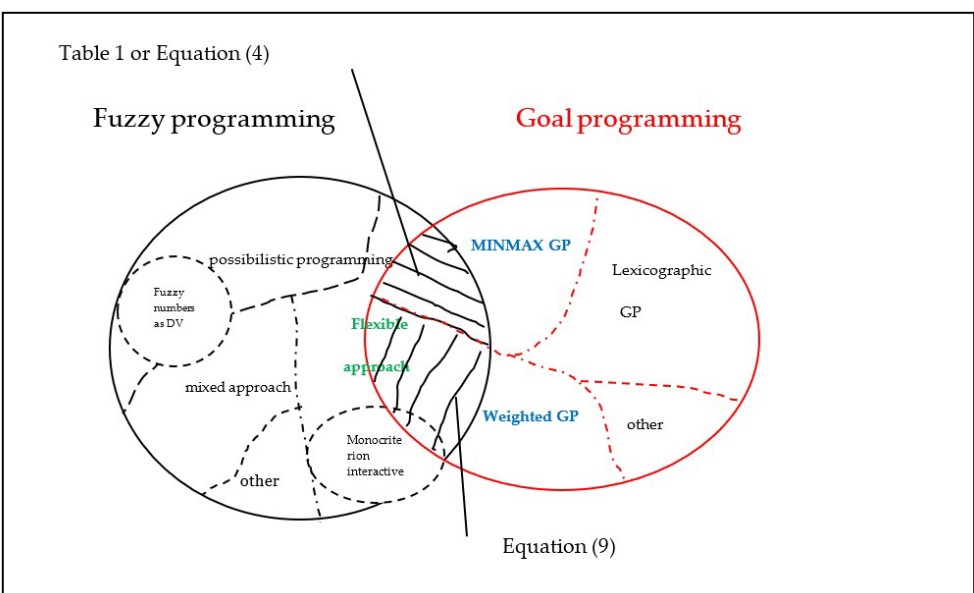

**Figure 3.** Schools in fuzzy programming and goal programming and the common approaches.

Mixed approaches combining the concept of membership function with the use of fuzzy inequalities can often be found [56]. Even if the fuzzy programming is widely used in multicriteria problems, it can be also used for one objective function with fuzzy constraints by following an interactive approach [57].

The flexible programming is widely used in reservoir management, but often with a monocriterion formulation [58]. The objective function mainly refers to the satisfaction of the demands or other quantitative approaches. Although the use of fuzzy parameters would be more realistic (as occurs in possibilistic programming), this assumption adds a significant computational complexity, since the simulation with fuzzy parameters includes problem sub-optimization based on the extension principle [59]. In this study, two objective functions are considered; the possibilistic programming formulation was avoided in order to achieve the coupling between the real simulation (e.g., Excel based) and the optimization model.

However, in this article, the authors deal with the flexible programming and its link with the GP (dashed area in Figure 2). It is obvious that GP and fuzzy programming are not identical, though the two approaches can be used in the case of flexible programming under the aforementioned approaches regarding the allowable diversions. Therefore, if the exceedance and the shortage from the ideal values are selected as the width (or tolerance) of the fuzzy constraints, then the flexible approach and the MINMAX GP are identical. Furthermore, it is proposed that these tolerances be established based on the pay-off matrix.

Hence the scientific mechanism consists of the following three steps:

1. Step 1: The water balance is modulated based on the examined problem according to needs and constraints.
2. Step 2: Two objective functions are examined: the maximum economic benefit, that is, the maximum irrigation areas, and the minimum retention time.
3. Step 3: Based on the previous solutions presented in the pay-off matrix (Table 2), the membership functions of the corresponding two goals are constructed.

**Table 2.** Pay-off matrix; the value of the objective functions are marked with grey.

| Goal | $x_3$ (m$^3$/month/hm$^3$) | $x_1 = A_1$ (stremmas/y) | $x_2$ (stremmas/y) | Retention Time Index = $\tau$ |
|---|---|---|---|---|
| Max($x_1$) | 2.61 | 73,977 | 24,659 | 229.72 |
| Min($\tau$) | 4.09 | 47,371 | 15,790 | 180.7 |

Finally, the third step concerns the calculation of the compromise solution through multi-criteria analysis and fuzzy sets and logic. The aforementioned two different goals are aggregated based on the min intersection, which is the intersection of the crisp sets (Equation (4), or equivalently, Table 1). Alternatively, the use of the weighted GP problem can be examined (Equation (9)). The is modulated under the condition that the corresponding acceptable ranges are modulated based on the pay-off matrix, that is, the grey zone of the decision (Equation (8)).

The authors highlight that based on the fuzzy flexible approach, not only the direct use of the weights is eliminated, but furthermore, an understandable solution is achieved. The decision maker can check the value of the achieved membership function for each objective and, if it is sufficient, then it can be accepted. In the next sections, the solution and the produced results are presented and discussed.

## 5. Results

Initially, the abstraction parameters are quantified. Due to the small contribution of the Karla basin's inflows, in the last years, only 12 hm$^3$ can be directed to satisfy agricultural needs. Even if the abstracted amount of water does not reflect an intensive agricultural use, the seasonality of abstractions contributes negatively to the reservoir's water quality, especially during warm seasons when the needs are intensified. The calculated agricultural needs, based on the adopted scenario, are presented in Figure 4.

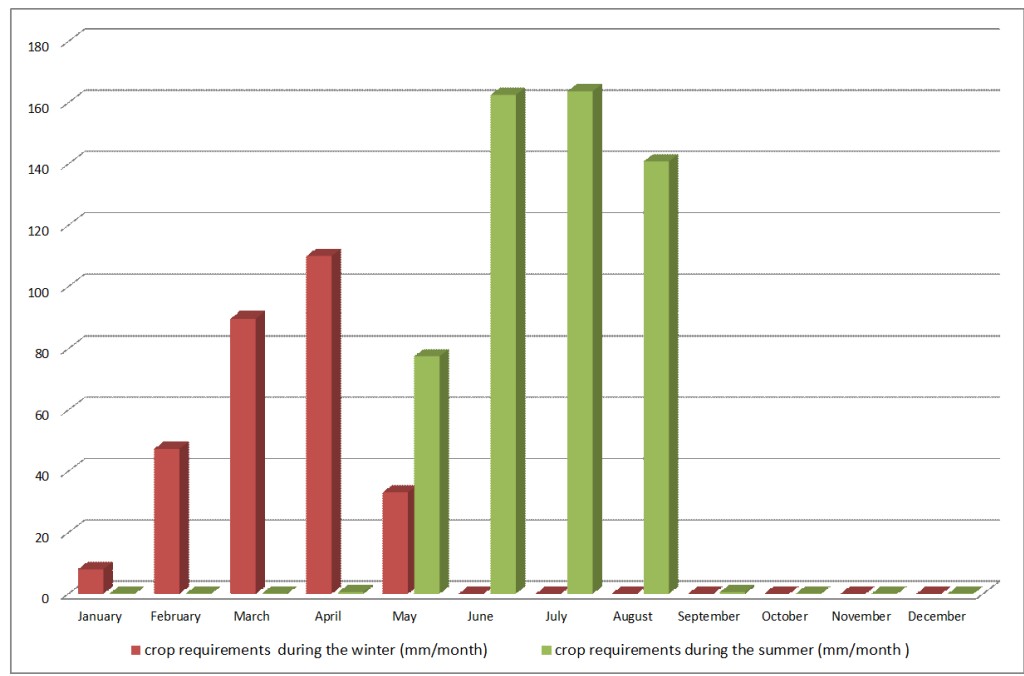

**Figure 4.** Monthly water requirements of both the current crops and the proposed winter crops.

The irrigation needs mentioned above, along with agricultural evapotranspiration, were included in the calculation of the reservoir's mass-balance with respect to the average hydrological year. Moreover, as such, for each month of this year, the extractions from the Pinios, the evaporation of Karla's reservoir, the rainfall affecting the reservoir, the releases of the reservoir, and the runoff from its drainage basin, among others, were specified. The contribution of the Karla basin is achieved by using the Giakoumakis et al. [43] model (calibrated for 2018 hydrological year).

Second, an individual monocriterion approach is established. Two objective functions are examined: the maximum economic benefit, that is, the maximum irrigation areas, and the minimum retention time. Based on these solutions, the membership functions of the corresponding two goals are constructed.

Finally, the third step concerns the calculation of the compromise solution through the use of a multi-criteria analysis and fuzzy sets and logic. The aforementioned two different goals are aggregated based on the min intersection. Finally, the problem concludes to a nonlinear programming problem, which is solved in Excel, and hence, a common degree of satisfaction of both goals is achieved [60].

Therefore, the methodology is implemented based on the three steps:

### 5.1. Step 1

The water balance was finally formed according to needs and constraints as follows:

$$V_i = V_{i-1} + Vi_{rainfall} + Vi_{outflow} + Vi_{Pinios} - Vi_{evap} - Vi_{crops1} - Vi_{crops2} - Vi_{leaks} - x_3 \quad (10)$$

where V is the volume of the reservoir, $Vi_{rainfall}$ is the precipitation over a lake surface, $Vi_{outflow}$ is the runoff to the reservoir from its drainage (local) basin, $Vi_{Pinios}$ is the monthly contribution of the Pinios river, $Vi_{evap}$ is the evaporation over a lake surface, $Vi_{leaks}$ is the leaks from the reservoir, $Vi_{crops1}$, $Vi_{crops2}$ are the water withdrawals for the summer and the winter corps, respectively, and $x_3$ is the additional water supply for many purposes.

The initial volume regarding the mean conditions must be equal to 65 hm$^3$ (near the minimum acceptable volume for ecological reasons). In any month, the water volume in the lake must be greater than or equal to 60 hm$^3$. No release is permitted regarding the mean conditions, and furthermore, a volume of 42.74 hm$^3$ remains for flood protection:

$$x_1 - 3x_2 = 0 \quad (11)$$

which is a relation between the summer and the winter crops.

$$60 \leq V_i, i = 1, \ldots, 12 \quad (12)$$

which indicates the required minimum volume for ecological purposes, allowing the lake to verify the ecological criteria as a wetland.

$$V_i + 42.74 \leq 183.88 \quad (13)$$

this constraint is for flood protection.

$$V_{13} = V_0 \quad (14)$$

Another critical point is the initial storage and the final storage. As Dutta [61] proposed, the final storage equals to the initial storage, and this means that the sum of all inflows is equal to the sum of outflows (including the evapotranspiration and the leakage, see Equation (14)).

It is evident that the initial storage must be greater than 60 hm$^3$ in order to satisfy Equation (12). On the other hand, with a significant evaporation, and by taking into account that the main inflow occurs in winter and the main outflow occurs in summer, a large amount of initial storage in our problem could lead to lower cultivated areas. The initial storage was considered to be 65 hm$^3$.

The decision variables are selected to be the area for the summer crops ($x_1$) and the winter crops ($x_2$) and the constant monthly resale, either to the municipality of Volos or for other uses, to improve the retention time ($x_3$). The problem is addressed by following the next steps.

### 5.2. Step 2

Two objective functions are examined: the maximum economic benefit. that is, the maximum irrigation areas, and the minimum retention time.

$$\text{Max } A_1 \quad (15)$$

s.t water balance constraints.

$$\min \left( \sum_{t=1}^{12} \frac{V_i}{Q_i} = \tau \right) \tag{16}$$

s.t water balance constraints.

The second objective function is an index to describe the goal of minimum retention time. $Q_i$ is the monthly total outflow from the reservoir, that is, the sum of the water agricultural supply and the water amount $x_3$.

*5.3. Step 3*

Based on these solutions, which are presented in the pay-off matrix (Table 2), the membership functions of the corresponding two goals are constructed. For simplicity reasons, a linear shape is adopted:

$$\mu_2(\boldsymbol{x}) = \frac{A_1 - 47,371}{73,977 - 47,371}, \tag{17}$$

$$\mu_1(\boldsymbol{x}) = 1 - \frac{(\tau - 180.7)}{229.7 - 180.7} \tag{18}$$

Finally, the third step concerns the calculation of the compromise solution through multi-criteria analysis and fuzzy sets and logic. The aforementioned two different goals are aggregated based on the min intersection, which is the intersection of the crisp sets. Finally, the problem concludes to a nonlinear programming problem, which is solved in Excel, and hence a common degree of satisfaction of both goals is achieved. The auxiliary variable $\lambda$ denotes the common satisfaction degree of both the objective functions:

$$\begin{cases} \max \lambda \\ \text{s.t.} \\ \mu_1(\boldsymbol{x}) \geq \alpha \ (f_1(\boldsymbol{x}) = \tau \widetilde{\leq} 180.7 \ cost) \\ \mu_2(\boldsymbol{x}) \geq \alpha \ (f_2(\boldsymbol{x}) = \boldsymbol{x}_1 \widetilde{\geq} 73,977 \ benefit) \\ \text{crisp constraints} \end{cases} \tag{19}$$

The above equation leads to $x_3 = 3.29$ hm$^3$/month, $x_1 = 61,814$ stremmas/y, $x_2 = 20,605$ stremmas/y, and $\tau = 203.14$. As aforementioned, a way to verify the results and their meaning is the value of the membership function. Here, a common level regarding the two objectives is achieved with $\lambda = 0.54$, which can be characterized as an acceptable solution. This value is significantly above zero, which in fuzzy sets is the lowest membership function value and above the neutral value of 0.5 (where the complement is equal with the membership function itself)). This solution can be characterized as acceptable, feasible, and sustainable.

## 6. Discussion

The environmental objectives for all European water bodies are defined in Article 4 of the WFD. The aim is long-term sustainable water management based on a high level of protection of the aquatic environment. Within that general objective, specific environmental objectives are defined for heavily modified water bodies, such as Lake Karla, i.e., good ecological potential and good chemical status by 2027 [7]. Basic measures for the efficient and sustainable use of water were planned for the entire river basin in the previous cycles of RBMPs, but most of them were never implemented; thus, the Commission's recommendations have been partially fulfilled.

Especially for lake Karla, the adopted Program of Measures (PoM) [39] necessitates maintaining lower acceptable ecological water levels. Another crucial measure is the undertaking of targeted actions for compliance with a continuous and minimal water supply throughout the year. Both are endorsed in the proposed solution of the present research. It is true that sustainability of water resources in agroecosystems is a complex issue, as it

depends on various interdependent aspects. To fully understand the interlinkages between the reservoir operation, the supported primary production (and the ecosystem services that can be offered), and the possible economic and ecological benefits, it is necessary to examine the balance of these targets and how they affect sustainability in a holistic way.

First, the calculation of the reservoir's mass-balance with respect to the average hydrological year revealed, besides a lack of quantitative data, that the current situation describes a not fully operating reservoir. As aforementioned, the hydrological regime of lake Karla has dramatically changed. Nowadays, it is clear that the estimated inflow of $35$ hm$^3$ coming from the basin of Karla was an overestimation in the construction plans. The main source is the Pinios river, without the surface runoff from Karla's subbasin. Based on the proposed solution, the target is almost 61,814 stremmas/y, with 20,605 irrigated from the lake, which is smaller than the initial target of 92,500 stremmas/y.

Managing the re-constructed lake Karla seems to be a difficult issue because of conflicting criteria features such as environmental standards, provisional ecosystem services, and economic needs. An interesting point is that, even if there is no initial consideration of the uncertainty, the fuzziness is introduced in the problem. Indeed, in Equation (16), the first membership function expresses a benefit criterion (maximum cultivated areas with summer crops); therefore, its goal can be seen as a fuzzy inequality constraint based on the previous solutions. The same holds for the cost objective function (minimum $\tau$). From a more mathematical point of view, the fuzziness arises from the conflict of goals. The direct use of weights is also eliminated by using the proposed method, and this can be seen as an advantage of the method.

The achieved solution is not computationally complicated, and the process is understandable by the decision makers. In fact, the direct use of the weights is avoided by using the membership function. In addition, the use of the min intersection leads to a commensurate solution. The fuzzy logic simulates the human way of thinking, and hence the use of the min intersection provides an environmentally friendly and understandable decision. More analytically, the use of the min intersection to integrate the two goals leads to a non-compensation between the two goals, leading to a common acceptable degree.

From a strictly mathematical point of view, things are quite different. The GP is a multi-objective programming model, which can be characterized as a distance-based method. In the era where the majority of the reviewers compared the fuzzy approach with the probabilistic approach, Hannan [52] investigated the relation between GP and fuzzy flexible programming. Yaghoobi and Tamiz [53] suggested that applying fuzzy set theory in GP has the advantage that the decision maker is allowed to specify imprecise aspiration levels. Hence, the fuzziness is used to interpret the inequalities:

$$A_1 \widetilde{\geq} 73,977 \tag{20}$$

$$\tau \widetilde{\leq} 180.7 \tag{21}$$

In the same way, the equality constraints can be expressed. Consequently, based on Hannan's [48] comments, Yaghoobi and Tamiz [53] proved that the flexible programming (here, Equation (16)) is equivalent to the following MINMAX GP problem [53,62]:

$$\begin{cases} \max \lambda \\ \tau - p \leq 180.7 \\ x_1 + n \geq 73,977 \\ \lambda + \frac{1}{\Delta^+}p \leq 1 \\ \lambda + \frac{1}{\Delta^-}n \leq 1 \\ n, p \geq 0 \text{ i} \\ X \in C_S \end{cases}, \tag{22}$$

where $\Delta^+ = 229.7 - 180.7$ is the maximum allowable distance of exceedance and $\Delta^- = 73,977 - 47,371$ is the maximum allowable distance of shortage. These values came

from the pay-off matrix. However, according to the conventional MINMAX GP, the values $\Delta^+, \Delta^-$ can be seen (from the view of GP) as weights. In other words, Equations (18) and (19) are identical. Hence, the fuzziness can provide an interpretable suggestion regarding the weights of the MINMAX GP problem. The tolerance of the fuzzy inequalities is based on the pay-off matrix. Hence, from another point of view, the use of the membership function is not something different from the use of the weights. Simply by using the concept of membership functions, the weights arise from the allowable tolerance of each constraint. Obviously, the decision maker could change the tolerance of the membership function in the case that he or she tries to change the solution. Because of the strong relation between the GP and the (fuzzy) flexible programming, some authors nowadays use the term fuzzy GP instead of GP [63]. In the case that the weighted model is used, then the model becomes

$$\min\left(w_2 \frac{p}{(229.7 - 180.7)} + w_1 \frac{n}{(73,977 - 47,371)}\right)$$

s.t.

$$\tau(\mathbf{x}) - p \le 180.7,$$
$$A_1(\mathbf{x}) + n \ge 73,977,$$
$$n, p \ge 0, v$$
$$\frac{n}{(73,977 - 47,371)} \le 1,$$
$$\frac{p}{(229.7 - 180.7)} \le 1,$$
$$X \in C_s$$

(23)

An interesting point is the behavior of the weights. If the decision maker selects: $w_2 = 0.4$, $w_1 = 0.6$, then, based on the previous Equation, the results are almost identical, with the monocriterion solution for 73,977 stremmas as maximum cultivated areas. Hence, the full compensation between the objectives leads sometimes to non–balanced solutions. However, the above weighted model could be useful for problems with multiple objective functions and the existent of sub-criteria with allowable compensation among them.

An interesting point for further review is the use of the intuitive fuzzy environment for fuzzy flexible programming problems. This approach would take into account the degrees of satisfaction and dissatisfaction of objectives, and hence the compensation between the score of the criteria could be reduced with the use of a weighted model [64]. However, based on the proposed solution, the multi-purpose nature of the lake is maintained, including water safety for floods, ecosystem services, agriculture, and protection and improvement of the aquifers, even if the hydrological regime is not convenient. This research complements the findings of other researchers.

The delay in construction works, the environmental violations, and the lack of environmental policy have been noted previously [33], but several issues still exist. In-lake ecological modeling showed that the retention time creates many problems, eutrophication and cyanobacteria included [38]; thus, our suggestion of preserving a constant inflow from the Pinios and an artificial outflow could be of aid in ameliorating ecological quality. Moreover, by reducing retention time, a nutrient seasonal pattern could return, because, until recently, high nutrient concentrations revealed historical sediment contamination of the lake [44].

Panagopoulos and Dimitriou [4] support, through their research, that the Karla reservoir is capable as a project to "combat water scarcity, achieving a twofold crop yield production and respective agricultural income in the surrounding area, securing the coverage of the water supply needs of the closest city, improving the status of groundwater resources, developing a natural shelter for biodiversity and emerging recreation and touristic opportunities". By including crop rotation (among others) in Arc-SWAT, 8000 ha were modeled to be irrigated exclusively by the lake, resulting in a seasonal water abstraction of 40 hm$^3$. They claim that only high flood flows of the Pinios are adequate to store enough water volume to assist the reservoir operation during low flow periods, preserving the

reservoir storage limit of 100 hm$^3$; however, this cannot support a constant outflow of the lake, minimizing (even artificially) water retention time. Another important point is that effective coverage of irrigation needs is only 43%, meaning that the solution of 1/3 crop rotation suggested in this research guarantees effectiveness and productivity.

The adoption of a new management (including hydroeconomics) approach to enhance water use efficiency is also suggested by various researchers [65–67]. Alamanos et al. [65,66] state that resource overexploitation forges high values for the unmet demand and a negative water balance, highlighting as a major problem the agricultural water supply and its relation to the aquifer depletion. The suggested solution in this research, the alteration in water provisioning to and from the reservoir, could satisfy the local needs and lower the resource and environmental costs, which are usually high in degraded areas [67].

## 7. Conclusions

This research deals with the restoration of lake Karla, pinpointing the major problem of water resource optimal use. The multi-purpose reservoir operation is an open problem, since the works have not been finished, and furthermore, the current practices demonstrate significant diversion from the initial design. These synthesize a multi-dimensional issue, with many aspects and complex interactions. For this purpose, fuzzy flexible programming was adjusted for the problem.

Flexible GP overcame subjectivity through the min intersection and maximized benefits for both goals and supported decision making in favor of the reservoir sustainability, based on the proposed scenario. The strong relation between the GP and the fuzzy flexible programming was presented, along with further interpretation of the membership function.

It is observed that the use of fuzzy sets and logic, as the membership function and min intersection, can provide decision makers with efficient and understandable tools to quantify the policy planning. The proposed solution takes into account the local economy and food production, the ecological services of the lake, the artificial recharge of the groundwater, and the soil improvement, among others, and hence, a sustainable integrated proposal is modulated.

By using the proposed method, unbalanced solutions can be avoided, and furthermore, the value of the membership function provides an understandable way to understand and check the solution. By using flexible programming, the direct use of the weights is avoided.

**Author Contributions:** Conceptualization, I.K. and M.S.; methodology, M.S. and L.V.; software, M.S.; validation, D.L. and M.S.; formal analysis, K.R., E.K. and D.L.; resources, L.V. and I.K.; data curation, K.R., E.K. and L.V.; writing—original draft preparation, K.R., E.K, I.K., L.V., D.L. and M.S.; writing—review and editing, I.K., D.L. and M.S.; visualization, L.V. and M.S.; supervision, I.K. and M.S. All authors have read and agreed to the published version of the manuscript.

**Funding:** This research received no external funding.

**Institutional Review Board Statement:** Not applicable.

**Informed Consent Statement:** Not applicable.

**Acknowledgments:** The authors would like to express their gratitude for the important of input, primary data, and scenario support of P. Sidiropoulos and the staff of the Lake Karla Management Body.

**Conflicts of Interest:** The authors declare no conflict of interest.

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
