# Peer review of "Flexible Goal Programming for Supporting Lake Karla’s (Greece) Sustainable Operation"

_sustainability, doi:10.3390/su14074311_

Round 1

Reviewer 1 Report

sustainability-1544714-peer-review-v1

 Flexible Goal Programming for Supporting Sustainable Operation in Lake Karla, Greece.

Thank you very much for inviting me as a reviewer for this article. In this study, authors describe a mathematical model flexible goal programming for supporting sustainable operation in Lake Karla, Greece. My comments are as follows.

  • The introduction directly jumps to the technical details. It should be written in such a way to let the readers clearly understand of each step of the study.
  • Page 1, L2, in Title, author mentions “flexible goal programming”. what is the different between flexible goal programming and conventional goal programming?
  • The manuscript is poorly structured. Why is a literature review jam-packed within the introduction and not presented as a standalone section?
  • In abstract, it is unclear that why the flexible goal programming is applied. Which problem that the flexible goal programming aims to model is also confusing. A flexible goal programming approach is applied to develop solution. but, what does the solution means? If the solution is for decision making, what are the final decisions.
  • Add a transition paragraph and describe every section as the last paragraph of introduction.
  • There is not smooth transition between the sections. It is not clear the relationship of sections 2 and 3 with 4.
  • The authors should correct the punctuations throughout the manuscript. For example at the end of equations (3), (4), (6),.. put dot.
  • The "Where" below Eq. (1) should be "where". Remove the similar problems in your paper.
  • In section 4, page 5, the proposed optimization model “Methodological Approach” is not described in a proper way. Author needs to describe this model in detail.
  • Problem statement section is too weak. You should discuss the problem in more detail and explain the main superiority of the proposed model.
  • Mathematical modelling is required to be fully explained. But in this paper, there isn't the mathematical modelling for optimization.
  • It is important to give graphical simulation for the solution.
  • There is no justification of the method. Why for this problem area, please discuss? There are many other similar methods in the literature in this area, so such a justification is required.
  • How could you set the parameters of the algorithm and model? How can you validate the values?
  • The key point of this model is not prominent enough, please explain the algorithm design part and algorithm comparison part in detail.
  • More description on Examples should be inserted in the paper.
  • The values of parameters could be a complicated problem itself, how the authors give the values of parameters.
  • The paper lacks the running environment, including software and hardware. Please supplement them. It is convenient for other researchers to redo your experiments and this makes your work easy acceptance.
  • The paper did not perform any sensitivity test. To check the feasibility and validation of the proposed approach over a wide range, the sensitivity test is recommended to conduct.
  • The paper did not compare the obtained results with the existing approaches. Without comparing the obtained results, how the author can claim the proposed approach a better approach? Author is suggested to give comparative of the proposed approach with other existing methods. All comparative results should be put in a table. For more clarity, the comparative study should be demonstrated with the help of graphs.
  • It would be better if the author adds some'managerial implications' in Result and Discussion section of this manuscript.
  • The conclusion of the article needs to be further refined and revised. The current conclusion does not highlight the value of the article, and it overlaps with the content in the abstract and introduction.
  • In addition, some minor typos and grammatical errors have been found at some parts. Please re-read this article again and correct all typos and errors.

Author Response

Reviewer #1 - Comments

The authors thank the reviewer for his/her constructive and useful comments. All comments made by the reviewer have been addressed and the mistakes have been corrected in the revised annotated paper (see revised manuscript and annotated manuscript). The reviewer comments are shown in “plain text” and the authors’ response in the “text with italics”.

  • The introduction directly jumps to the technical details. It should be written in such a way to let the readers clearly understand of each step of the study.
  • Thank you for this comment. The authors agree with the reviewer and the comment has been addressed in the revised manuscript. The Introduction is completely rewritten including new elements and/or sections according to reviewer guidelines.  The transition became smoother, explanatory sentences were added and the introduction is followed by the study area, the problem definition, the aim and then the methodology (see Introduction section of the revised manuscript).
  • Page 1, L2, in Title, author mentions “flexible goal programming”. what is the different between flexible goal programming and conventional goal programming?
  • The authors thank the reviewer for his/her constructive and useful comment. We agree with the reviewer that a clearer description was needed in the manuscript and the comment has been addressed in the revised manuscript. Annotation was added in the introduction section and the distinctions/similarities between the goal programming and the flexible programming is analytically and graphically now presented in the new revised manuscript in methodology section ‘4.2 Fuzzy flexible programming’. Furthermore, two (2) equations (8&9), 1 table and a new graph were added in the revised manuscript. Please see Section 4.2 (about 1,5 pages)

 “An interesting point of view […] it can be also used for one objective function with fuzzy constraints by following an interactive approach.”

  • The manuscript is poorly structured. Why is a literature review jam-packed within the introduction and not presented as a standalone section?
  • The authors partially agree with the reviewer and the comment has been partly addressed in the revised manuscript. The authors believe that is usual practice to include part of the literature review in the Introduction section. Specifically, the literature review in the Introduction section is a brief description of the state of the art, and it was included as a standalone part within the Introduction section since our aim was not to review the existing literature. Furthermore, in each section the discussion is based also in comparison with recent studies from the international literature (see also revised manuscript).
  • In abstract, it is unclear that why the flexible goal programming is applied. Which problem that the flexible goal programming aims to model is also confusing. A flexible goal programming approach is applied to develop solution. but, what does the solution means? If the solution is for decision making, what are the final decisions.
  • We thank the reviewer for this comment. Indeed, these points are addressed in the revised manuscript. The abstract was rewritten with respect to the restriction of 200 words (please see Abstract in the revised manuscript).
  • Add a transition paragraph and describe every section as the last paragraph of introduction.
  • The authors agree with the reviewer and the comment has been addressed in the revised manuscript. A sentence was added at the end of each section and a short description of the manuscript layout was also included the introduction:

“The paper is organized as follows […] presented and discussed”

  • There is not smooth transition between the sections. It is not clear the relationship of sections 2 and 3 with 4.
  • As stated before, A sentence was added at the end of each section for a more proper transition (please see the revised manuscript)
  • The authors should correct the punctuations throughout the manuscript. For example at the end of equations (3), (4), (6),.. put dot.
  • The authors agree with the reviewer and the comment has been addressed in the revised manuscript.
  • The "Where" below Eq. (1) should be "where". Remove the similar problems in your paper
  • The entire manuscript was reviewed by a native speaker for grammar and syntax errors. It is accepted if the equation stops with a comma to proceed to the term’s explanation with lower sentence letters. So, when the "where" is referred to the previous equation, we put comma and thereafter not capital letter is used.
  • In section 4, page 5, the proposed optimization model “Methodological Approach” is not described in a proper way. Author needs to describe this model in detail.
  • The authors partially agree with the reviewer and the comment has been partly addressed in the revised manuscript. Since this sub section contained well known knowledge, we deemed there was no need for more detailed description due to paper length limitations. However, in the revised manuscript scientific references are provided for the interested readers. Furthermore, It should be mentioned that in the results section a more analytical presentation of the examined hydrological budget was made in the revised manuscript.
  • Problem statement section is too weak. You should discuss the problem in more detail and explain the main superiority of the proposed model.
  • The authors agree with the reviewer and the comment has been addressed in the revised manuscript. All major problems were mentioned. Though some further analysis was made in the in the respective sections of the revised manuscript.
  • Mathematical modelling is required to be fully explained. But in this paper, there isn't the mathematical modelling for optimization.
  • The authors agree with the reviewer and the comment has been addressed in the revised manuscript. Indeed, so, after all other inputs, the optimization problem is now analytically explained in the revised 4.2. section and in the last sections in the revised version.
  • It is important to give graphical simulation for the solution.
  • We understand the reviewer’s comment but, in our opinion a graphical simulation of the solution would confuse the readers than facilitate the understanding. Please, keep in mind that we came up with a solution in a problem with more than three decision variables, so these complicated multidimensional schemes are not informative to engineers.
  • There is no justification of the method. Why for this problem area, please discuss? There are many other similar methods in the literature in this area, so such a justification is required.
  • We thank the reviewer for this comment since we initially though it was optimal to present the selected methodology than to explain the method selection. So in the revised version the selection of the optimization model was done. The flexible programming is firstly compared with other approaches and then selected to be used. These is justified in the section of the mathematical description and by taking into account the results. The reviewer can find in the 4.2. section:

“Τhe flexible programming is widely used in  […]the real simulation (e.g. Excel based) and the optimization model.”

A comparison between the weighted model and the flexible programming can be found also in the section of discussion and equation 23 was added for its compliment:

“In the case that the weighted model is used then the model becomes: […] existent of sub-criteria with allowable compensation between them.”

  • How could you set the parameters of the algorithm and model? How can you validate the values?
  • Information of the real operation of the reservoir is included in the model to satisfy the validity of the applied parameter ranges. Hence, validation is not a prerequisite in the use of such models (see for example Mendoza and Ventura, 2012). Though, the solution is checked based on the values of the membership functions and an explanatory sentence was added in the revised manuscript:

“The above equation leads to […] This solution can be characterized as feasible and sustainable. “

  • The key point of this model is not prominent enough, please explain the algorithm design part and algorithm comparison part in detail.
  • The authors thank the reviewer for the comment and this comment has been addressed in section 4.2 of the revised manuscript. In addition, the complete algorithm is described in steps:

“Hence the scientific mechanism consists of the following three steps […] then it can be accepted. In the next sections the results are presented and discussed.”

  • More description on Examples should be inserted in the paper.
  • The authors in all sections have provided international references of similar studies. The bibliography section now includes 67 references in the revised manuscript (14 additionally references have been included in the 1st version of the manuscript). Furthermore, this study deals with a specific problem, in a complicated case study where no similar examples can be inserted and compared using goa programming techniques.
  • The values of parameters could be a complicated problem itself, how the authors give the values of parameters.
  • The authors agree with the reviewer and special attention was given for the parameter feasibility. In this study, the parameters are selected based on the hydrological standards, international directives and national official documents. Furthermore, the key question regarding the fuzzy tolerance is presented analytically including the corresponding methodology (equations 6-9) for the pay-off matrix, the lowest acceptable (tolerance) and the aspiration levels, weights and membership functions:

“By using linear membership functions the key question is the selection of their thresholds […]

  • The paper lacks the running environment, including software and hardware. Please supplement them. It is convenient for other researchers to redo your experiments and this makes your work easy acceptance.
  • The solution was given in an excel spreadsheet. Description of the procedure is fully given. As for hardware simple laptop or PC equipped with a spreadsheet included in an office suite (e.g. Microsoft Office) is the only requirement.
  • The paper did not perform any sensitivity test. To check the feasibility and validation of the proposed approach over a wide range, the sensitivity test is recommended to conduct.
  • The authors totally agree with the reviewer. However, the proposed method is an approach tailor- made for this exact case study so, given the dataset at hand, it would be wrong to arbitrary alter the timeseries and parameters of the solution to test the sensitivity. Though, an explanation was added in the revised manuscript and equation 23 of the revised manuscript:

“In the case that the weighted model is used then the model becomes […]

  • The paper did not compare the obtained results with the existing approaches. Without comparing the obtained results, how the author can claim the proposed approach a better approach? Author is suggested to give comparative of the proposed approach with other existing methods. All comparative results should be put in a table. For more clarity, the comparative study should be demonstrated with the help of graphs
  • The economic benefits deriving from agricultural water use and environmental protection are based on the minimum intersection. For this purpose, flexible (fuzzy) programming is used and compared with MINMAX goal programming model although the weights are not used directly as in the flexible fuzzy programming. An understandable compromise solution (the maximum economic benefit from irrigation areas and the minimization of water retention time) is achieved and furthermore, the values of the membership functions can be used to verify the solution. The proposed solution leads to a quantitative proposition, incorporating new findings from modeling the recent real operation of the reservoir and is compared with the weighted approach and the monocriterion approach in the revised manuscript.
  • It would be better if the author adds some'managerial implications' in Result and Discussion section of this manuscript.
  • The authors agree with the reviewer and the comment has been addressed in the revised manuscript. The managemental applications of this approach are presented and highlighted in the Results and Discussion sections of the revised manuscript.
  • The conclusion of the article needs to be further refined and revised. The current conclusion does not highlight the value of the article, and it overlaps with the content in the abstract and introduction.
  • The authors agree with the reviewer and the comment has been addressed in the revised manuscript. It was ameliorated and the following lines were added:

“It is observed that the use of fuzzy sets and logic […] programming the direct use of the weights is avoided.”

  • In addition, some minor typos and grammatical errors have been found at some parts. Please re-read this article again and correct all typos and errors.
  • As stated before, the entire manuscript was reviewed by a native speaker for grammar and syntax errors

  1. Mendoza, A.; Ventura, J.A. Analytical Models for Supplier Selection and Order Quantity Allocation. Applied Mathematical Modelling 2012, 36, 3826–3835, doi:1016/j.apm.2011.11.025.

Reviewer 2 Report

line 32: „requires” because the logical subject is ensuring, not water systems.

line 314, in the table 1, the measure unit of the last but one column is half in English half in Greek, it is probably acre/season. By the way, the why do you use acre instead of hectares?

line 321: crisp sets instead of crisp logic

Author Response

Reviewer #2 - Comments

The authors thank the reviewer for his/her constructive and useful comments. All comments made by the reviewer have been addressed and the mistakes have been corrected in the revised annotated paper (see revised manuscript and annotated manuscript). The reviewer comments are shown in “plain text” and the authors’ response in the “text with italics”.

  • line 32: „requires” because the logical subject is ensuring, not water systems.
  • line 314, in the table 1, the measure unit of the last but one column is half in English half in Greek, it is probably acre/season. By the way, the why do you use acre instead of hectares?
  • line 321: crisp sets instead of crisp logic
  • We thank the reviewer for his comments. All points were addressed and corrected. We should note that acres were changed to stremmas (1000m2) since this is the most usual unit in Greece, included in the SI.  

Reviewer 3 Report

This manuscript by author provides an interesting Flexible Goal Programming for Supporting Sustainable Operation in Lake Karla, Greece, and I can say the same things about the manuscript.

Detailed comments:

  1. What is innovation of this M.S., author should give the more innovation clearly in introduction part? And there are many similar researches, what is the news?

  1. This study aimed to develop a new method to estimate flexible Goal Programming for Supporting Sustainable Operation in Lake Karla, Greece, what is the flexible Goal Programming, authors should give the clear description, should clarify the specific connotation and indicators.

  1. The research objective is to achieve a compromise solution with respect to both the economic benefits deriving from agricultural water use and environmental protection, what is the specific premise and boundary?

  1. The problem is addressed through nonlinear programming and a common degree of goal satisfaction is achieved supporting sustain-ability and water use optimization

, what is scientific mechanism? And How to verify?

  1. Two objective functions/goals were examined (the maximum economic benefit from irrigation areas and the minimization of water retention time) and membership functions of the corresponding goals were constructed, Are there any qualitative problems?

  1. About the Flexible Goal Programming, Do you need to consider future development and changes?

Author Response

Reviewer #3 - Comments

The authors thank the reviewer for his/her constructive and useful comments. All comments made by the reviewer have been addressed and the mistakes have been corrected in the revised annotated paper (see revised manuscript and annotated manuscript). The reviewer comments are shown in “plain text” and the authors’ response in the “text with italics”.

  • This manuscript by author provides an interesting Flexible Goal Programming for Supporting Sustainable Operation in Lake Karla, Greece, and I can say the same things about the manuscript.
  • We thank the reviewer for his kind general comment.
  • What is innovation of this M.S., author should give the more innovation clearly in introduction part? And there are many similar reseaches, what is the news?
  • The authors agree with the reviewer and the comment has been addressed in the revised manuscript. It was indeed missing and in the revised version of the manuscript is clarified in the Introduction section:

“The article deals with […] proposed (fuzzy) multicriteria approach an analytical quantitative solution is achieved and not some general strategies.”

Moreover, several points were added in the entire manuscript highlighting the innovation and practical value of this approach in the revised manuscript. The Introduction section is completely rewritten including new elements and/or sections according to reviewer guidelines. Furthermore, the inclusion of new elements based on the reservoir operation as highlighted in the Introduction section that clearly states the innovation of the proposed method (Lines 121-139 of the revised manuscript).

“The article deals with the restored Lake Karla …, we highlight that based on the proposed (fuzzy) multicriteria approach an analytical quantitative solution is achieved and not some general strategies.”

  • This study aimed to develop a new method to estimate flexible Goal Programming for Supporting Sustainable Operation in Lake Karla, Greece, what is the flexible Goal Programming, authors should give the clear description, should clarify the specific connotation and indicators.
  • The authors thank the reviewer for his/her constructive and useful comment. We agree with the reviewer that a clearer description was needed in the manuscript and the comment has been addressed in the revised manuscript. Annotation was added in the introduction section and the distinctions/similarities between the goal programming and the flexible programming is analytically and graphically now presented in the new revised manuscript in methodology section ‘4.2 Fuzzy flexible programming’. Furthermore, two (2) equations (8&9), 1 table and a new graph were added in the revised manuscript. Please see Section 4.2 (about 1,5 pages)

“An interesting point of view […] it can be also used for one objective function with fuzzy constraints by following an interactive approach.  

  • The research objective is to achieve a compromise solution with respect to both the economic benefits deriving from agricultural water use and environmental protection, what is the specific premise and boundary?
  • The key question regarding the fuzzy tolerance is presented analytically including the corresponding methodology (equations 6-9) for the pay-off matrix, the lowest acceptable (tolerance) and the aspiration levels, weights and membership functions:

“By using linear membership functions the key question is the selection of their thresholds […]

Furthermore, all hypotheses and admissions concerning input data before the parametrization were based on official national documents, European guidance and case specific literature. Section 4.2. of the revised manuscript describes analytically the employed method with all the necessary information.

  • The problem is addressed through nonlinear programming and a common degree of goal satisfaction is achieved supporting sustain-ability and water use optimization, what is scientific mechanism? And How to verify?
  • These questions are adressed in the revised section 4.2. In addition, all the algorithm is described in steps:

“Hence the scientific mechanism consists of the following three steps […] then it can be accepted. In the next sections the results are presented and discussed.”

As for the verification some paragraphs were added:

“The authors highlight that based on the fuzzy flexible approach […] In the next sections the results are presented and discussed”

And in Results question

“The above equation leads to x3= 3.29 hm3/month x1 = 61,814 stremmas/y, x2 =20,605 stremmas/y and τ = 203.14. Αs aforementioned […] This solution can be characterized as feasible and sustainable.”

  • Two objective functions/goals were examined (the maximum economic benefit from irrigation areas and the minimization of water retention time) and membership functions of the corresponding goals were constructed, Are there any qualitative problems?
  • Qualitative problems are described in the second and third bullet of 3.1 section and an explanation on the relation among retention time and water quality is further given in the revised manuscript.
  • About the Flexible Goal Programming, Do you need to consider future development and changes?
  • It should be clarified that the flexible programming in our case study is preferred among other methods of optimization It was done in the new manuscript:

“An interesting point for further review is the use of the intuitionistic fuzzy environment upon the fuzzy flexible programming problems. Hence, this approach will take into account the degrees of satisfaction and dissatisfaction of objectives”

By using possibilistic programming (fuzzy parameters) the problems becomes more complicated

“Τhe flexible programming is widely used in reservoir problem bur mainly with a monocriterion formulation […] couple between the real simulation (e.g. Excel based) and the optimization model.”

More sentences on that were also added in results and discussion sections of the revised manuscript.

Reviewer 4 Report

The manuscript is well written. However, lack of comparison between the new findings and previous finding can be seen. Improve the discussion section by highlighting the importance of your finding compare with the previous research works. 

Author Response

Reviewer #4 - Comments

The authors thank the reviewer for his/her constructive and useful comments. All comments made by the reviewer have been addressed and the mistakes have been corrected in the revised annotated paper (see revised manuscript and annotated manuscript). The reviewer comments are shown in “plain text” and the authors’ response in the “text with italics”.

  • The manuscript is well written. However, lack of comparison between the new findings and previous finding can be seen. Improve the discussion section by highlighting the importance of your finding compare with the previous research works. 
  • We thank the reviewer for his kind words and this valuable comment. I was indeed missing and a paragraph was added in the Introduction section of the revised manuscript. Moreover, several points were added in the entire manuscript highlighting the innovation and practical value of this approach in the revised manuscript. The Introduction section is completely rewritten including new elements and/or sections according to reviewer guidelines. Furthermore, the inclusion of new elements based on the reservoir operation as highlighted in the Introduction section that clearly states the innovation of the proposed method (Lines 121-139 of the revised manuscript).

“The article deals with the restored Lake Karla …, we highlight that based on the proposed (fuzzy) multicriteria approach an analytical quantitative solution is achieved and not some general strategies.”

  • In addition, in the Discussion section the results are compared with previous studies in the region and the outcomes of this study are used for operational water resources management practices.
  • Finally, the mathematical tool is compared with similar models with more clarity:

“Μore analytically in section 4.2 it is presented that under some hypothesis about the acceptable range, the (flexible) fuzzy programming is identical with the MINMAX goal programming model although the weights are not be used when the fuzzy flexible programming is applied.”

The mathematical details are provided in both Section 4.2. and in the Discussion section.

Round 2

Reviewer 1 Report

The authors' responses to the referee's comments are incomplete, confusing, and incomprehensible.